# Fault Diagnosis Method for Aircraft EHA Based on FCNN and MSPSO Hyperparameter Optimization

Xudong Li, Yanjun Li *, Yuyuan Cao, Shixuan Duan, Xingye Wang and Zejian Zhao

College of Civil Aviation, Nanjing University of Aeronautics and Astronautics, Nanjing 210016, China
* Correspondence: lyj@nuaa.edu.cn

**Featured Application: An intelligent fault diagnosis method based on a convolutional neural network and a particle swarm optimization algorithm is proposed for the troubleshooting of the aircraft EHA.**

**Abstract:** Contrapose the highly integrated, multiple types of faults and complex working conditions of aircraft electro hydrostatic actuator (EHA), to effectively identify its typical faults, we propose a fault diagnosis method based on fusion convolutional neural networks (FCNN). First, the aircraft EHA fault data is encoded by gram angle difference field (GADF) to obtain the fault feature images. Then we build a FCNN model that integrates the 1DCNN and 2DCNN, where the original 1D fault data is the input of the 1DCNN model, and the feature images obtained by GADF transformation are used as the input of 2DCNN. Multiple convolution and pooling operations are performed on each of these inputs to extract the features. Next these feature vectors are spliced in the convergence layer, and the fully connected layers and the Softmax layers are finally used to attain the classification of aircraft EHA faults. Furthermore, the multi-strategy hybrid particle swarm optimization (MSPSO) algorithm is applied to optimize the FCNN to obtain a better combination of FCNN hyperparameters; MSPSO incorporates various strategies, including an initialization strategy based on homogenization and randomization, and an adaptive inertia weighting strategy, etc. The experimental result indicates that the FCNN model optimized by MSPSO achieves an accuracy of 96.86% for identifying typical faults of the aircraft EHA, respectively, higher than the 1DCNN and the 2DCNN by about 16.5% and 5.7%. By comparing with LeNet-5, GoogleNet, AlexNet, and GRU, the FCNN model presents the highest diagnostic accuracy, less time in training and testing. The comprehensive performance of the proposed model is demonstrated to be much stronger. Additionally, the FCNN model improved by MSPSO has a higher accuracy rate when compared to PSO.

**Keywords:** electro hydrostatic actuator; fusion convolutional neural networks; particle swarm optimization; gram angle difference field

## 1. Introduction

EHA is an actuator of aircraft rudder surface based on the PBW(power-by-wire) actuator system; its appearance makes the hydraulic system of conventional aircraft simpler and enhances the aircraft's structure and performance [1,2]. The basic principle of EHA is volume control, it means the motor or pump directly controls the flow and direction of oil flowing through the actuator cylinder. Depending on whether the regulating object is a motor or a pump, or both, EHA can be divided into three categories [3]: fixed displacement variable speed (EHA-FPVS), variable displacement fixed speed (EHA-VPFS), and variable displacement variable speed (EHA-VPVS). As a complex electromechanical system, EHA is highly integrated with a power control unit, electronic control unit, variable-speed motor, piston pump, check valve, bypass valve, relief valve, actuator cylinder, sensor, and other modules, which complicates its faults, including control unit failure, sensor failure (loss of

sensor gain), motor circuit failure (short circuit, open circuit, and resistance increases) and hydraulic circuit failure (hydraulic oil pollution, internal leakage), etc.

As a future development trend of the aircraft actuation system, how to develop a reliable and effective fault diagnosis method based on the fault signature of EHA has drawn the interest of many academics in recent years. Ma, elaborated on the composition and working principle of EHA, described its mathematical model, and provided the theoretical basis for establishing the failure model [4]. With the technology for fault prediction and health management, Xu concentrated on studying the EHA intelligent fault diagnosis and prediction method [5]. Muhammad Haq Nawaz proposed an EHA fault detection and isolation analysis method based on the bond graph [6]. Matteo D. L. Dalla Vedova uses genetic algorithms to monitor the performance degradation of EHA [7]. Liu Jun applied the gray correlation analysis to the fault diagnosis of EHA [8]. Cui studied the failure injection of EHA and its effect on aircraft flight performance [9]. Based on the adaptive neural network robust observer, Zhao created an EHA fault diagnosis and fault tolerance controller, which successfully improved EHA's robustness to faults [10]. S. Andrew Gadsden investigates a mathematical model for EHA fault diagnosis and proposes a new model-based fault diagnosis strategy [11].

However, the above-mentioned fault diagnosis methods for EHA are mostly model-based, and there are problems that the linear models established are too large, causing system instability, and the models cannot clearly express the deep logical relationships of the system. As a result, these methods have a poor ability to identify faults, a high rate of false alarms, and poor generalization. Nevertheless, with significant developments in sensors, big data, and artificial intelligence technology, data-based intelligent fault diagnosis methods have been developing rapidly in recent years. They no longer focus on the object's physical structure but analyze the data to find the relationship between fault data and fault mode. When failures occur, the fault mode can be quickly identified with fault data, which significantly improves the efficiency of fault diagnosis. Especially for intricate electromechanical systems like the aircraft EHA, collecting the operational fault data and establishing a data-based fault diagnosis model has great practical value in improving the efficiency of fault diagnosis.

As a popular tool in data-based fault diagnosis, the convolutional neural network has performed well in recent years and benefits from its sparsity, shared weights, and other advantages. Yang proposed a hybrid fault diagnosis algorithm combining a one-dimensional convolutional neural network and support vector machine, which shows high precision in fault diagnosis of rolling bearings [12]. Li studied the application of CNN in fault diagnosis of aircraft hydraulic systems, the 1DMCCNN model proposed by Li realized the processing of one-dimensional time series signals and multi-sensor fusion [13]. Ji combined the hydraulic cylinder pressure signals into a two-dimensional matrix and fed it into CNN to diagnose the leakage fault of hydraulic cylinders [14]. Zhang encoded the one-dimensional bearing vibration signals into the time-series images and then used them as input for two-dimensional CNN for fault diagnosis. The result shows that the encoded images have a stronger ability to express the original fault features, which gives full play to the advantages of two-dimensional CNN in image processing [15].

But, without the proper selection of hyperparameters, such as the number of filters, types of activation function, training batch size, learning rate, etc., CNN cannot perform well in the fault diagnosis. Usually, people adjust the hyperparameters manually based on experience and through constant trial and error, but new CNN tends to have more layers, leading to a surge in the number of hyperparameters. Therefore, it is almost impossible to pinpoint a close-to-optimal hyper-parameter configuration for a CNN manually under a reasonable cost, which hampers the adaption of CNNs for various real-world problems [15]. Hence, several CNN optimization methods for hyper-parameter selection have been proposed in recent years, including the particle swarm optimization, a well-known SI algorithm, which has drawn much interest from researchers for optimizing CNN

and other neural networks. The iteration rate and learning effectiveness of CNN can be increased through structural optimization of CNN using the PSO algorithm [16,17].

The studies indicate that CNN, as a data-based fault diagnosis method, can make up for the shortcomings of traditional fault diagnosis methods. Moreover, the 1DCNN can process sequence data effectively, while the 2DCNN has more advantages in image processing. Therefore, we propose a fault diagnosis method based on FCNN for the fault diagnosis of the aircraft EHA. FCNN combines the fault features of two dimensions, enriches the fault feature data of various dimensions, and makes it possible to efficiently extract fault features from numerous system characteristic parameters of aircraft EHA. Moreover, the MSPSO is employed to optimize the FCNN model's hyperparameters. The MSPSO algorithm combines multiple strategies; compared with PSO, a better combination of FCNN hyperparameters can be obtained.

The FCNN model and MSPSO algorithm are expounded in Sections 2 and 3. In Section 4, the process of MSPSO optimizing the hyperparameters of the FCNN model is mainly introduced. Section 5 describes how to obtain the fault data of the aircraft EHA through the EHA-FPVS model built-in lab, and how to construct fault samples. In Section 6, the proposed MSPSO-FCNN model is compared with other models, and the compare results are analyzed and discussed. The conclusion is summarized in Section 7.

## 2. Fusion Convolutional Neural Network (FCNN)

### 2.1. Two-Dimensional Convolutional Neural Network (2DCNN)

The 2DCNN is a traditional convolutional neural network. It was originally designed as a deep learning method for image processing and recognition. It can be used to extract two-dimensional signal structure features as well as significant features from images [17].

A typical two-dimensional convolutional neural network consists of input layer, convolutional layer, pooling layer, fully connected layer, and output layer. The output layer is generally used to solve the regression and classification problems through Sigmoid or Softmax activation functions [18], and the structure of 2DCNN is displayed in Figure 1.

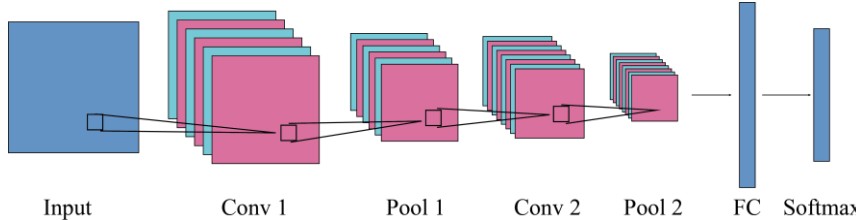

Input          Conv 1          Pool 1          Conv 2          Pool 2          FC    Softmax

**Figure 1.** Structure of 2DCNN model.

### 2.1.1. Convolutional Layer

The convolution layer is made up of multiple two-dimensional feature planes. As the core module of the CNN, it is connected to other layers through sparse connections [19,20]. The mathematical expression of the convolution operation process can be described by:

$$x_j^l = f\left(\sum_{i=1}^{M} x_i^{l-1} * k_{ij}^l + b_j^l\right), j = 1, 2, \cdots N \tag{1}$$

where, $x_j^l$ is the feature graph; $k_{ij}^l$ is the weight matrix of the convolution kernel; $b_j^l$ is the bias; $f$ is the nonlinear activation function; $l$ is the $l_{th}$ layer of the network; $M$ is the number of the feature graph; and $N$ is the number of the convolution kernel.

The two-dimensional convolution operation diagram is shown in Figure 2. The size of the input feature graph is $5 \times 5$, the convolution kernel size is $5 \times 5$, and the step size is 2.

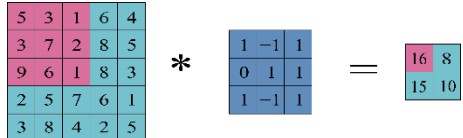

**Figure 2.** Convolution operation of 2DCNN.

### 2.1.2. Pooling Layer

The pooling layer is also called the down-sampling layer and is usually placed after the convolution layer. The pooling layer is divided into maximum-pooling and average-pooling, it can reduce the mesh parameters and calculated amount while retaining the main features to control the risk of overfitting [20]. Maximum-pooling and average-pooling are defined as follows:

$$P_{i.\max}^{l+1}(j) = \max_{(j-1)w+1 \le t \le iw} \left[ q_t^l(t) \right] \tag{2}$$

$$P_{i.avg}^{l+1}(j) = \frac{1}{w} \max_{i=(j-1)w+1} \left[ q_t^l(t) \right] \tag{3}$$

where, $w$ is the receptive field; $q_t^l(t)$ is the input of the neuron and $P_i^{l+1}$ is the output.

### 2.1.3. Fully Connected Layer

The fully connected layer is behind the last pooling layer. Each neuron in the fully connected layer is connected to all feature maps in the last pooling layer, and the advanced features in the fully connected layer are extracted as the input of the classifier [21]. The mathematical model of the fully connected layer is described as follows:

$$F_j^l = \sum_{i=1}^n x_i^{l-1} * w_{ij}^l + b_j^l \tag{4}$$

where, $F_j^l$ is the output of the neuron; $w_{ij}^l$ and $b_j^l$ are the weights and biases, respectively.

### 2.2. One-Dimensional Convolutional Neural Network (1DCNN)

The 1DCNN is an improvement of the traditional 2DCNN. It can effectively extract essential features from sequence data and has apparent advantages in identifying simple patterns in data. After the one-dimensional signal is input into the network model, the features are automatically extracted layer by layer, and the extracted features' abstraction gradually becomes higher. Then the extracted features pass through the full connection layer and input layer, to realize the classification of different signals [12,22].

The principle of the 1DCNN processing data is to treat one-dimensional data as an image with a height of 1 pixel. The schematic diagram of the one-dimensional convolution operation is shown in Figure 3.

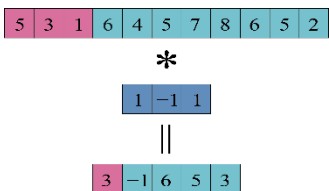

**Figure 3.** Convolution operation of 1DCNN.

The structure of the 1DCNN is similar to the 2DCNN, which also includes convolution layer, pooling layer, and full connection layer. Figure 4 shows multiple convolutional layers and pooling layers are stacked to receive one-dimensional input data and extract deep features from these data. Data features are continuously condensed by the convolution

kernel to obtain deep features. The pooling operation reduces the calculated quantity on the network.

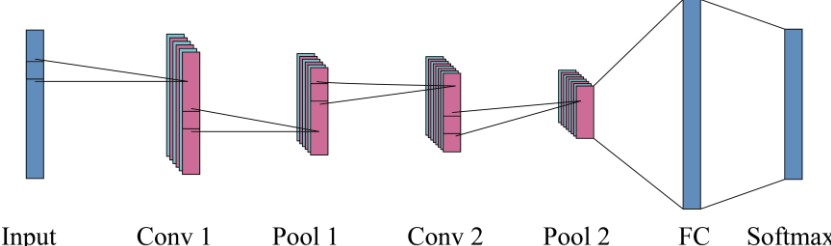

**Figure 4.** Structure of 1DCNN model.

*2.3. Multi-Feature Fusion Convolutional Neural Network (FCNN)*

During flight, the data collected by sensors are one-dimensional time-series signals with many original features, while CNN has more advantages in 2D image processing. Therefore, to fuse the features of the two dimensions, we propose a multi-feature fusion convolutional neural network combining 1DCNN and 2DCNN. Compared with ordinary CNN networks, the FCNN uses a network model with an additional convergence layer, which connects feature vectors of two dimensions to achieve the purpose of feature fusion. Additionally, a dropout layer is included to prevent FCNN from overfitting. Overfitting often occurs when training a neural network, which makes the diagnostic accuracy of the model in the training set much higher than that in the testing set. And a dropout layer can effectively alleviate the occurrence of such overfitting and play the regularization effect to a certain extent.

Figure 5 demonstrates the structure of the multi-feature fusion CNN model for aircraft EHA fault diagnosis proposed in this paper.

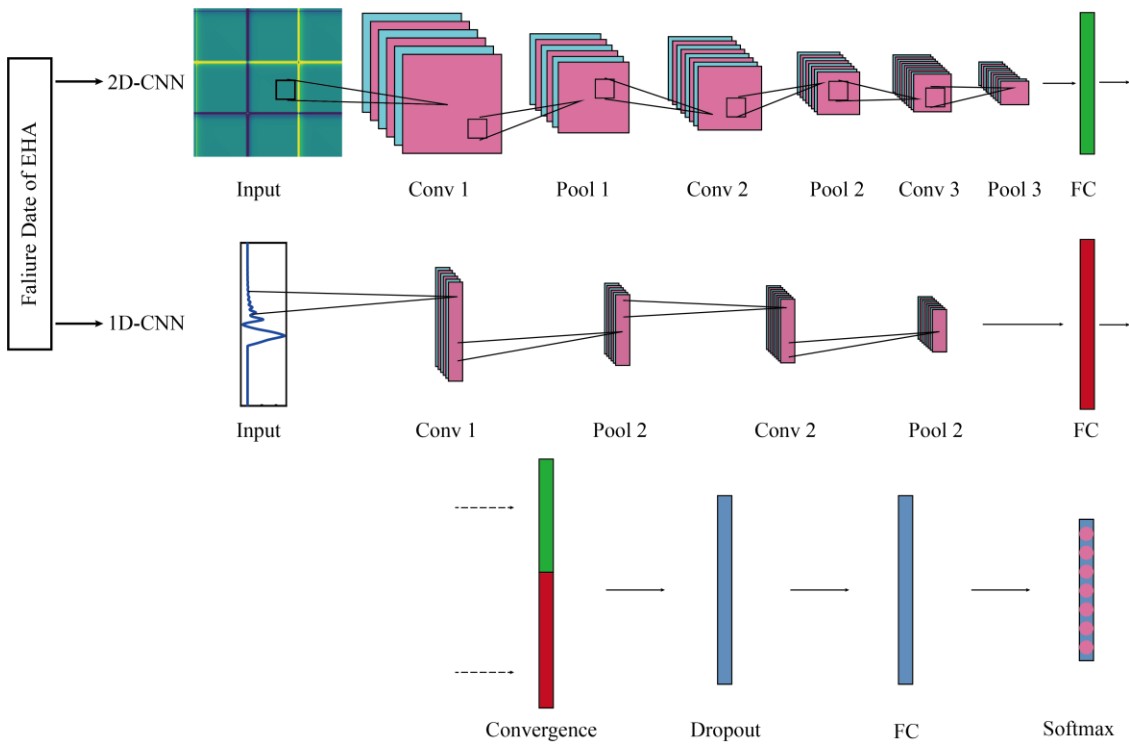

**Figure 5.** Structure of the proposed FCNN model.

## 3. Multi-Strategy Hybrid Particle Swarm Optimization Algorithm (MSPSO)

In the PSO algorithm, the behavior of particles is a kind of cooperative symbiosis. The search behavior of each particle is affected by other particles in the group, and the particle

itself will save the best position it has experienced. For each particle, it can be regarded as a candidate solution to the problem to be optimized, and its fitness value can be confirmed by the fitness function. The quality of the particle is judged by the fitness value [23,24].

### 3.1. Standard Particle Swarm Optimization Algorithm

Assuming that in a D-dimensional target search space, a population is made up of N particles, the $i_{th}$ particle represents a D-dimensional vector [25]:

$$x_i = (x_{i1}, x_{i2}, \cdots x_{iD}), i = 1, 2, \cdots N \tag{5}$$

And the velocity of the $i_{th}$ particle is also a D-dimensional vector, denoted as:

$$v_i = (v_{i1}, v_{i2}, \cdots v_{iD}), i = 1, 2, \cdots N \tag{6}$$

Each particle adjusts its flight velocity and path according to its experience and population experience to get closer to the best position. They evaluate their own fitness based on the objective function or fitness function. When the $i_{th}$ particle of generation T evolves to generation T + 1, it updates its velocity and position according to the following formulas:

$$v_i(t+1) = \omega v_i(t) + c_1 r_1(t) \left[ Pbest_i^t - x_i(t) \right] + c_2 r_2(t) \left[ Gbest^t - x_i(t) \right] \tag{7}$$

$$x_i(t+1) = x_i(t) + v_i(t+1) \tag{8}$$

In Formula (7), $\omega$ is the inertia weight; $c_1$ and $c_2$ are learning coefficients; $r_1$, $r_2$ denote random values from a uniform distribution in the range of [0, 1]; $Pbest_i$ is the personal best for the $i_{th}$ particle and $Gbest^t$ is the global best, and their values are updated according to the following formulas:

$$Pbest_i^{t+1} = \begin{cases} x_i^{t+1}, f\left(x_i^{t+1}\right) \geq f\left(Pbest_i^t\right) \\ Pbest_i^t, otherwise \end{cases} \tag{9}$$

$$Gbest^{t+1} = \underset{Pbest}{\operatorname{argmax}} \left[ f\left(Pbest_i^{t+1}\right) \right] \tag{10}$$

where $f(x_i)$ is the individual fitness value.

### 3.2. Multi-Strategy Hybrid Particle Swarm Optimization Algorithm

3.2.1. Initialization Strategy Based on Homogenization and Randomization

Population diversity plays an important role in improving the global search ability of the PSO algorithm and preventing the algorithm from falling into local optimum. The initial populations with diversity improve the algorithm's global search capability, while the currently commonly used population random initial strategy cannot guarantee a better coverage of the entire decision space [26]. Therefore, we adopt an initialization strategy based on uniformization and randomization, which can make the particles more uniformly distributed in the decision space while ensuring random initialization. The following describes how to initialize the population:

Step 1: Input the population size $n$, the decision vector dimension $d$, and the interval $\left[a_j, b_j\right]$ for each decision variable $x_j (j \in \{1, 2, \cdots d\})$;

Step 2: Divide the interval of decision variable $x_j$ into $n$ equal parts, that is, $\Delta j = (b_j - a_j)/n$;

Step 3: Define set $\Omega_j = \left\{ \left[a_j, a_j + \Delta j\right], \left[a_j + \Delta j, a_j + 2\Delta j\right], \cdots, \left[a_j + (n-1)\Delta j\right], b_j \right\}$;

Step 4: Select a random interval in $\Omega_j$ and assign a random value to $x_{ij} (i \in \{1, 2, \cdots n\})$ within that interval;

Step 5: Update collection $\Omega_j$: Remove the selected subinterval from the set $\Omega_i$ in step 4;

Step 6: Repeat steps 4–5 to obtain all values of the decision variable $x_j$: $(x_{1j}, x_{2j}, \cdots x_{nj})$;

Step 7: Repeat steps 2–6 for the final output of the initial population: $\{x_1, x_2, \cdots, x_n\}$.

### 3.2.2. Adaptive Inertia Weights and Learning Factor Strategies

The inertia weight $\omega$ is the most crucial parameter in the PSO algorithm, which balances the local search capability of the algorithm with the global search capability. The value of $\omega$ needs to be kept large in the early iterations to obtain a strong global search capability and search velocity. In contrast, a smaller value of $\omega$ is necessary for the later iterations to give the algorithm a higher local search capability and accuracy [27,28]. Therefore, we adopt a strategy where the inertia weights change adaptively with the iteration of the algorithm.

The basic idea of adaptive inertia weight is to adjust the inertia weights based on one or more feedback parameters, which can fully use the valuable information provided by the algorithm compared to the traditional constant inertia weight strategy and linear inertia weight strategy. We adaptively adjust the inertia weights using changes in the particle fitness values. Define the relatively change rate of particle fitness as follows:

$$k_i(t) = \begin{cases} \frac{f(x_i^t) - f(x_i^{t-1})}{f(x_i^{t-1})} & t \geq 2 \\ 0 & t = 1 \end{cases} \tag{11}$$

When $k_i(t)$ is large, it means that the particle is far away from the optimal solution, and the inertia weight should be increased to bring it closer to the optimal solution; while when $k_i(t)$ is small, it means that the particle is poorly updated or already attached to the optimal solution, and the inertia weight should be reduced to increase the particle local search ability. Based on this, the following inertia weight adjustment strategy is proposed:

$$\omega_i(t) = \frac{1}{1 + \exp(-k_i(t))} \tag{12}$$

In the PSO algorithm, the learning factors $c_1$ and $c_2$ determine the influence of the particle's experience and the population's experience on the particle motion trajectory, and setting larger or smaller values of $c_1$, $c_2$ is detrimental to the algorithm [29]. We let the learning factor vary with the inertia weight to obtain the appropriate $c_1$ and $c_2$. $c_1$ takes a larger value when the inertia weight is larger, and $c_2$ takes a smaller value, allowing the particles to lean more towards the individual optimum; while the opposite is true for small inertial weight, enabling the particles to move more toward the optimum population. The relationship among $c_1$, $c_2$, and $\omega$ is as follows:

$$\begin{cases} c_1 = \frac{1.7 \cos[\omega_i(t)]}{\omega_i(t)} - 0.4 \\ c_2 = 3.3\omega_i(t) \sin[\omega_i(t)] - 0.3 \end{cases} \tag{13}$$

Figure 6 shows how $c_1$, $c_2$ varies with $\omega$.

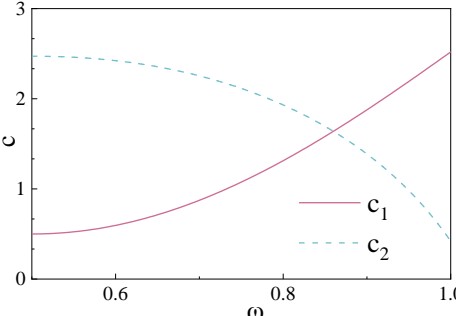

**Figure 6.** Evolution of $c_1$, $c_2$ as a function of $\omega$.

### 3.2.3. Hybrid Variation Strategy

As the algorithm iterates, many particles begin to aggregate and the particle population diversity decreases sharply, making the convergence velocity significantly slower in the later stages [30]. When the population diversity is too low, it is difficult for the algorithm to perform a normal operation. At this time, introducing a hybrid variation operator to increase the population diversity can promote further algorithm convergence.

First, the diversity of the population was judged based on the mean centroid distance of the particles [31]:

$$\text{Div} = \frac{1}{n} \sum_{i=1}^{n} \sqrt{\sum_{1}^{d} \left(x_{ij} - \bar{x}_j\right)^2} \tag{14}$$

where Div is the mean centroid distance of the particles, $\bar{x}_j = \frac{1}{n} \sum_{i=1}^{n} x_{ij}$ is the average value of all particles in the $j_{th}$ dimension. A smaller value of Div indicates a high aggregative of the particles around the center. Adopting the hybrid variation strategy when Div decays to a certain value, i.e.,

$$\text{Div}(t) \leq \alpha \text{Div}(0) \tag{15}$$

where $\text{Div}(t)$ is the mean centroid distance in the $t_{th}$ generation, $\text{Div}(0)$ is the initial mean center distance and $\alpha$ is the threshold coefficient. The literature [32] shows that when the population diversity during the iterative process decreases to 1–5% of the initial population, the search capability of particles decreases, and the algorithm tends to fall into a local optimum. Therefore, in practical applications, the threshold coefficient is generally set to 0.01–0.05, considering convergence accuracy and velocity.

After that, according to the variation rate $p_h$, randomly select $np_h$ particles in the original population for hybrid operation. The variation rate satisfies the following conditions:

$$p_h = p_{h0} \left(1 - \frac{t}{T}\right)^2 \tag{16}$$

where $p_{h0}$ is the initial variation rate and $T$ is the maximum number of iterations.

The child particles generated by the hybridization operation are deposited in the hybridization pool, and the positions and velocities of the offspring are generated by random crossover of the parents according to the following formula.

$$\begin{cases} x_i^* = p_b x_i + (1 - p_b) x_j \\ x_j^* = p_b x_j + (1 - p_b) x_i \\ v_i^* = \frac{v_i + v_j}{|v_i + v_j|} |v_i| \\ v_j^* = \frac{v_i + v_j}{|v_i + v_j|} |v_j| \end{cases} \tag{17}$$

where $p_b$ is a random number among [0, 1], $x_i^*, x_j^*, v_i^*, v_j^*$ are the position and velocity of the parents respectively; $x_i, x_j, v_i, v_j$ are the position and velocity of the children respectively.

Finally, the parent particles are replaced with child particles to form a new population, and the new population will continue to iterate.

### 3.2.4. Improved Handling Method of Boundary-Crossing Particles

Usually, at the end of each iteration, it is necessary to determine whether the particle is out of bounds. And particles out of bounds need to be adjusted. There are four methods to deal with particles out of bounds: absorbing, reflecting, invisible, and damping. Among them, absorbing and reflecting are more commonly used. However, these existing methods discard the potentially excellent search direction of the particles, which is not conducive to the search of the boundary region and decrease the diversity of the populations [33,34].

In this regard, we propose an improved boundary treatment method: When the particle generated by Formulas (1) and (2) is beyond the $j_{th}$ dimensional decision variable

boundary in the decision space, the inertia weight $\omega$ or one of the learning factors $c_1$, $c_2$ (depending on which one contributes more to the particle velocity in the $j_{th}$ dimensional component) is reduced to half. Then the position of the particle is recalculated, and if the new particle is still beyond the boundary, the above steps continue to be repeated until the particle can effectively fall within the boundary.

Figure 7 shows that when a particle crosses the boundary of a decision variable, $\omega$ is reduced by half so that the particle falls within the boundary. The improved boundary handling method shortens the particle search step by decreasing the values of $\omega$, $c_1$ or $c_2$, and induces the particles to seek optimum in a smaller region close to the boundary, improving the algorithm's convergence and accuracy.

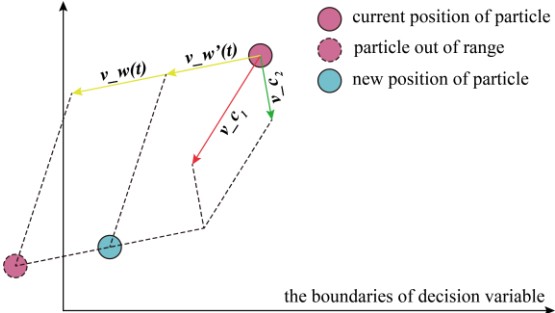

**Figure 7.** Improved particle boundary processing method.

3.2.5. Algorithm Flow

Based on the above four strategies, the steps of the MSPSO algorithm proposed in this paper are as follows (Algorithm 1):

---

**Algorithm 1:** Multi-Strategy Hybrid Particle Swarm Optimization Algorithm

---

1 : Generate $n$ particles, **initialize** the particle positions using the initialization strategy in Section 3.2.1, and calculate the values of $Pbest_i$ and $Gbest$ in the initial population.
2 : **while** $t < T_{\max}$
3 : Adjust the values of $\omega$ and $c_1$, $c_2$ according to Equations (12) and (13).
4 : Update particle positions and velocities according to Equations (7) and (8).
5 : Calculate the particle fitness values and update $Pbest_i$ and $Gbest$ according to Equations (9) and (10).
6 : **if** the particles out of bounds **then**
7 : Use the strategy in Section 3.2.4 to make the particles out of bounds fall within the boundary
8 : **end if**
9 : **if** the hybridization variation condition is satisfied according to Equations (14) and (15) **then**
10 : Generate a new population according to the hybrid mutation strategy of Equation (17).
11 : **end if**
12 : **end while**
13 : **output** the current optimal particle: $Gbest$

---

## 4. Fault Diagnosis Algorithm Based on MSPSO and FCNN

### 4.1. Structural Analysis of FCNN

The FCNN network model proposed in Section 2 contains 1DCNN and 2DCNN. The structure of 2DCNN is: input layer-convolutional layer ($C_1^2$)—pooling layer ($S_1^2$)—convolutional layer ($C_2^2$)—pooling layer ($S_2^2$)—convolutional layer ($C_3^2$)—pooling layer ($S_3^2$)—fully connected layer ($F^2$). The structure of 1DCNN is: input layer—convolutional layer ($C_1^1$)—pooling layer ($S_1^1$)—convolutional layer ($C_2^1$)—pooling layer ($S_2^1$)—fully connected layer ($F^1$). To fuse and classify the features extracted from the two dimensions of convolutional neural networks, the FCNN also contains: convergence layer—dropout layer—fully connected layer ($F$)—Softmax.

In FCNN, apart from parameters that require training like weights and biases, it also contains some hyperparameters that cannot be obtained by training. These hyperparameters include the filter quantity, filter size, filter step size, activation function, loss function type, learning rate, etc.

FCNN reduces the number of parameters during network training by combining three techniques: local sensing, weight sharing, and pooling, but some hyperparameters still need to be set manually, and the choice of these hyperparameters directly determines the structure of FCNN, thus affecting the performance of the network [35]. Due to the experimental environment, the step size of all convolutional layer filters is set to 1. The FCNN hyperparameters are shown in Table 1.

**Table 1.** Hyperparameters of FCNN.

| Particle | Hyperparameters | Particle | Hyperparameters |
|:---:|:---:|:---:|:---:|
| $x_1$ | Filter quantity of $C_1^2$ | $x_{11}$ | Activation function of $C_1^2$, $C_2^2$, $C_3^2$ |
| $x_2$ | Filter quantity of $C_2^2$ | $x_{12}$ | Activation function of $C_1^1$, $C_1^1$ |
| $x_3$ | Filter quantity of $C_3^2$ | $x_{13}$ | Activation function of $F^1$, $F^2$ |
| $x_4$ | Filter quantity of $C_1^1$ | $x_{14}$ | Size of $F$ |
| $x_5$ | Filter quantity of $C_2^1$ | $x_{15}$ | Activation function of $F$ |
| $x_6$ | Filter size of $C_1^2$ | $x_{16}$ | Optimizer |
| $x_7$ | Filter size of $C_2^2$ | $x_{17}$ | Learning rate |
| $x_8$ | Filter size of $C_3^2$ | $x_{18}$ | Dropout |
| $x_9$ | Filter size of $C_1^1$ | $x_{19}$ | Batchsize |
| $x_{10}$ | Filter size of $C_2^1$ | | |

### 4.2. Optimization of FCNN Structure Based on MSPSO

The structure of FCNN is very complex, and its performance is heavily dependent on the setting of hyperparameters. The hyperparameters of FCNN determine its structure, so the optimization of FCNN structure and the optimization of FCNN hyperparameters are essentially the same. The main objective of this paper is to maximize the accuracy of the FCNN model, which can be expressed as a function of the FCNN hyperparameters as follows:

$$Acc = f(x_1, x_2, \cdots x_{19}) \tag{18}$$

FCNN has numerous hyperparameters, and it is not easy to select them manually, while the PSO algorithm is simple, with few adjustable parameters and fast convergence, so the MSPSO algorithm proposed in Section 3 will be used to optimize the hyperparameters in FCNN. The hyperparameters in Table 1 will be used as each component of the particle's position in the high-dimensional space, and each particle represents one of the FCNN structures. The recognition accuracy is chosen as the value of the fitness function, and the hyperparameters of the FCNN model are automatically optimized via MSPSO.

### 4.3. The Process of MSPSO-FCNN Fault Diagnosis Method

The process of proposed MSPSO-FCNN fault diagnosis method for aircraft EHA fault diagnosis is as follows:

First, obtain the EHA failure sample dataset and encode the original one-dimensional data signals as two-dimensional feature images by GADF. After that, the data signals and feature images are input into 1DCNN and 2DCNN of the FCNN model, respectively. Then their features are extracted by multiple convolutional and pooling layers. These features are stretched into feature vectors by a fully connected layer and subsequently fused in the convergence layer and sent to the Softmax layer for classification after the fully connected layer. The accuracy of the fault classification will be used as the output of the MSPSO-FCNN model.

The position information of each particle in MSPSO will represent a combination of hyperparameters in FCNN, and the hyperparameters in FCNN will be set according to the decoding results of the particle position information. The fault classification accuracy

output from FCNN will be an input to MSPSO as the adaptation value of corresponding particles. MSPSO will get a new particle population after an iteration, reset the hyperparameters in FCNN, and constantly circulate until the optimal hyperparameter combination is obtained.

Finally, the data in the test set are input to the completed training model for fault diagnosis and output the fault diagnosis results. The specific flow is shown in Figure 8.

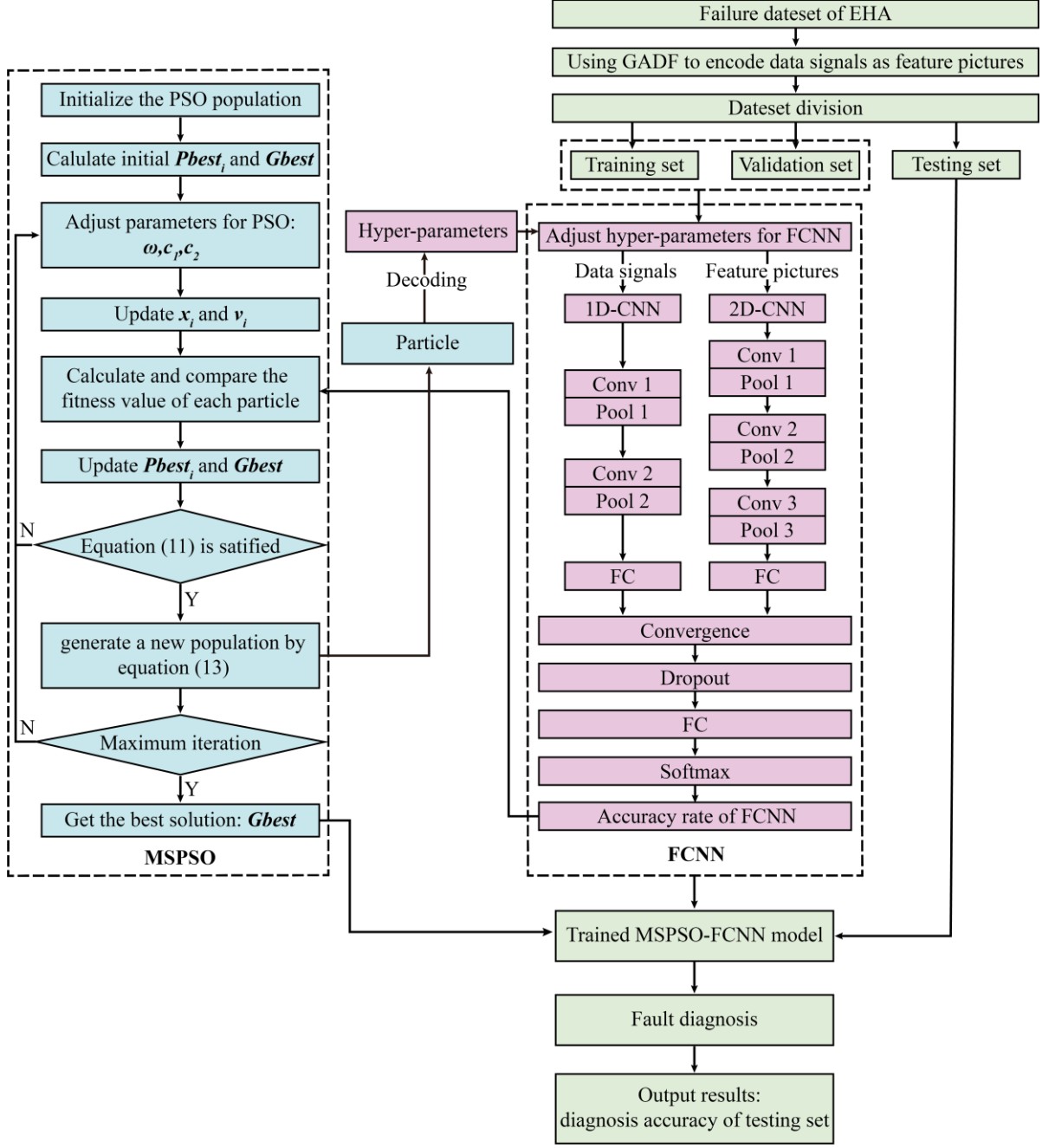

**Figure 8.** The framework of the MSPSO-FCNN fault diagnosis model.

## 5. Typical Fault Simulation and Fault Data Acquisition of EHA

### 5.1. Overview of the EHA Principle

Figure 9 shows the schematic diagram and picture of EHA-FPVS. The EHA comprises the control unit (including power control unit and electronic control unit), speed motor,

high-speed bi-directional pump, accumulator, check valve, bypass valve, safety valve, actuator cylinder, sensor, and other modules [3].

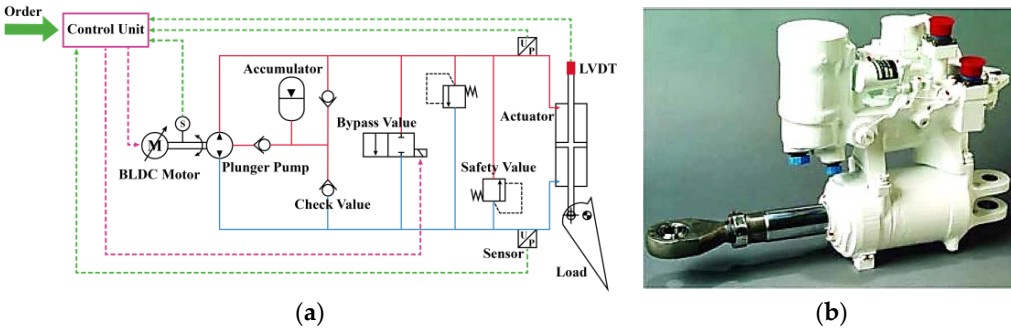

**(a)**    **(b)**

**Figure 9.** (**a**) Schematic diagram of EHA-FPVS; (**b**) Picture of EHA-FPVS.

When the control unit receives a command, it integrates sensor information such as actuator cylinder position, pressure, and rotatation speed to control the speed motor, which can quickly realize the volumetric control of oil flow in the hydraulic circuit. The high-speed pump is usually a bi-directional fixed displacement external gear pump or piston pump, which outputs high-pressure oil to one chamber of the actuator cylinder, resulting in a pressure difference between the two chambers and thus pushing the piston rod to overcome the rudder surface load to move. The accumulators mainly replenish hydraulic oil for the hydraulic circuit and provide a certain pressure for the pump's suction port to prevent the generation of air bubbles. The check valves are used to restrict fluid flow from the accumulator, and the safety valve protects the oil line and prevents excessive oil pressure from damaging the component. The safety valve protects the hydraulic circuit and prevents excessive oil pressure from damaging components. The bypass valve is used to achieve zero-load operation of the EHA or to conduct both ends of the pump directly to protect the hydraulic pump and motor in the event of a system failure.

*5.2. Typical Failure Simulation of EHA*

In this paper, a laboratory-built EHA-FPVS physical simulation model (Figure 10) is used for fault simulation, and the Supplementary Materials shows the model parameters of EHA-FPVS. Firstly, we obtained the data of EHA in normal operation mode and then injected the typical fault of EHA into the model to get the fault data. Six parameters characterizing the system performance were selected, including actuator cylinder displacement response, actuator cylinder velocity response, hydraulic pump pressure difference between two ports, hydraulic pump flow, motor speed, and accumulator outlet pressure, and the model diagnosed the EHA fault mode by learning the characteristics of these parameters.

In the simulation, a displacement command of 2 s duration was given continuously to the actuator cylinder within 16 s, and the maximum action range of the actuator cylinder was ±0.25 m. Figure 11a–c show the actuator cylinder displacement response, actuator cylinder velocity response, and motor speed under normal circumstances, respectively.

Select six typical faults of EHA, including internal leakage of the hydraulic pump, oil mingled with air, internal leakage of the actuator cylinder, increased friction between the actuator cylinder piston and the cylinder body, decreased sensor gain, and increased motor winding resistance. Set the corresponding parameters to inject these faults into the EHA:

(1) Internal leakage of the hydraulic pump: The clearance between the plunger and the cylinder was set to 0.1 mm, 0.2 mm, 0.3 mm, and 0.4 mm to simulate different degrees of internal leakage of the hydraulic pump.

(2) Oil mingled with air: The hydraulic oil air content is set to 2%, 3%, 4% and 5% to simulate different levels of air content.

(3) Internal leakage of the actuator cylinder: The clearance between the piston and the barrel was set to 0.1 mm, 0.2 mm, 0.3 mm, and 0.4 mm to simulate different degrees of failure.

(4)   Increased friction between the actuator cylinder piston and the cylinder body: The Coulomb friction between the piston and the barrel was set to 1.1 times, 1.2 times, 1.3 times, and 1.4 times of the normal condition to simulate different degrees of friction increase.

(5)   Decreased sensor gain: The displacement feedback loop gain was set to 0.9, 0.8, 0.7, and 0.6 to simulate different levels of gain reduction.

(6)   Increased motor winding resistance: Set the winding resistance to 3.6 Ω, 4.6 Ω, 5.6 Ω, 6.6 Ω to simulate different levels of failure.

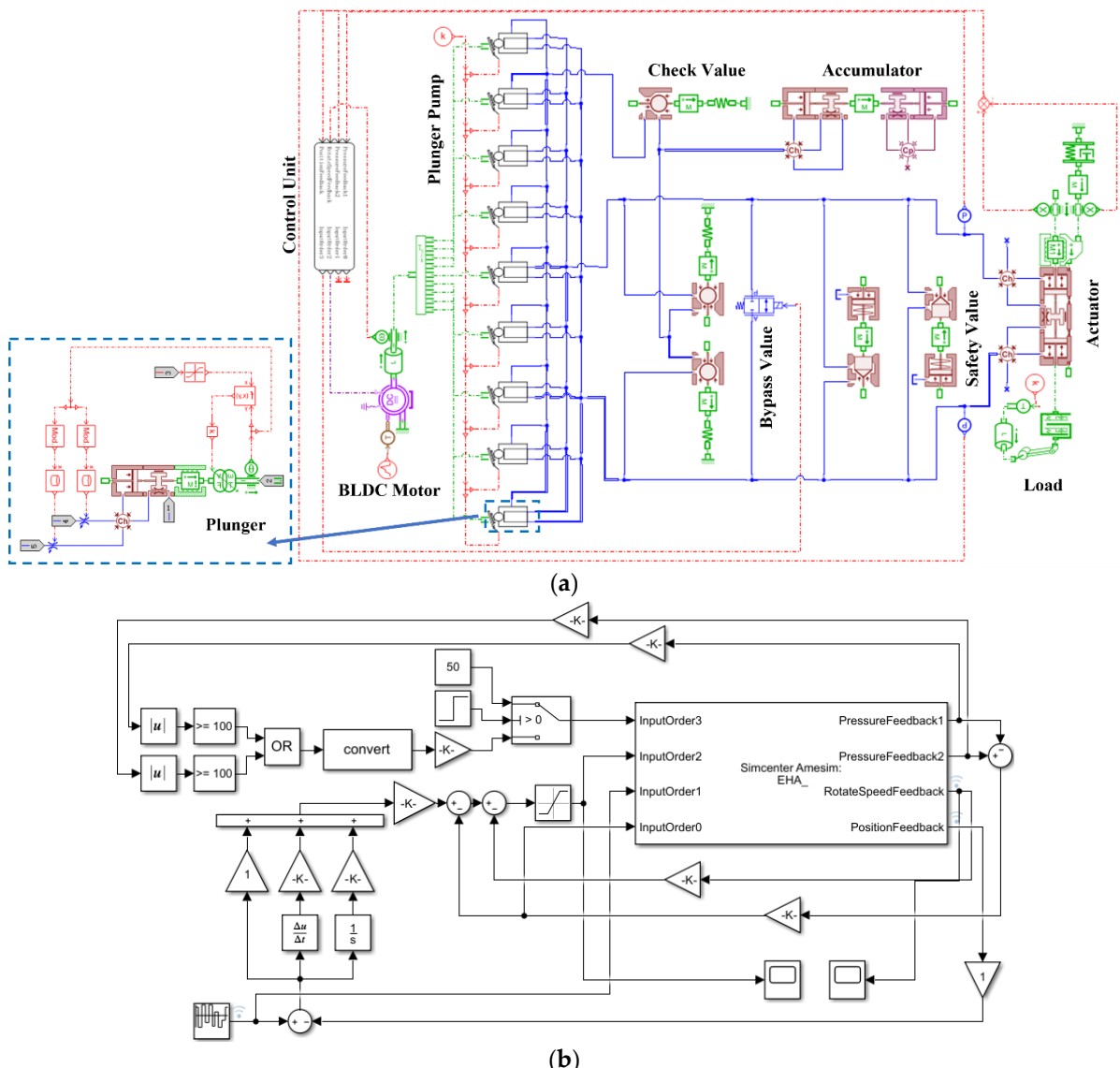

**Figure 10.** Physical simulation model of EHA-FPVS: (**a**) Model of hydraulic circuit; (**b**) Model of control unit.

Figure 12 shows the speed response curves of the actuator cylinder under different faults.

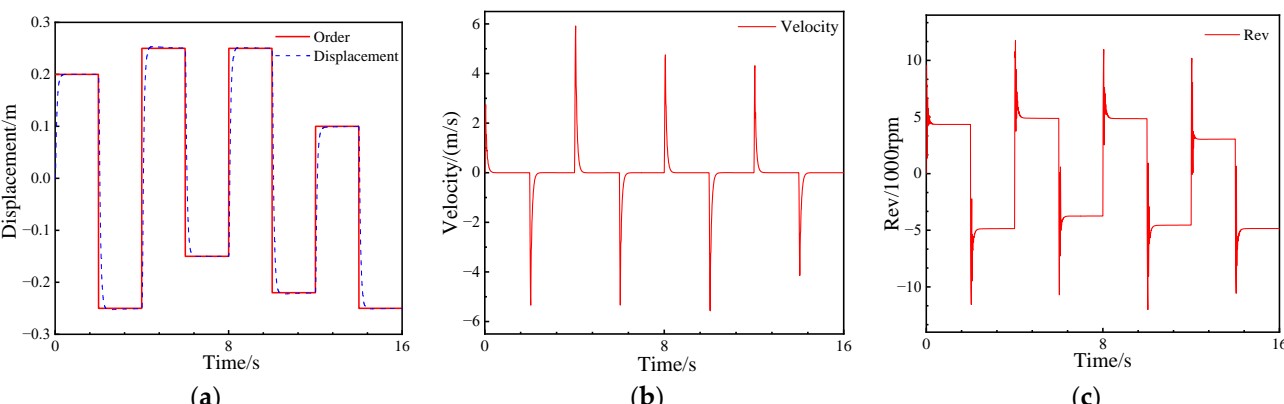

**Figure 11.** System response under normal circumstances: (**a**) actuator cylinder displacement response; (**b**) actuator cylinder velocity response; (**c**) rotation speed.

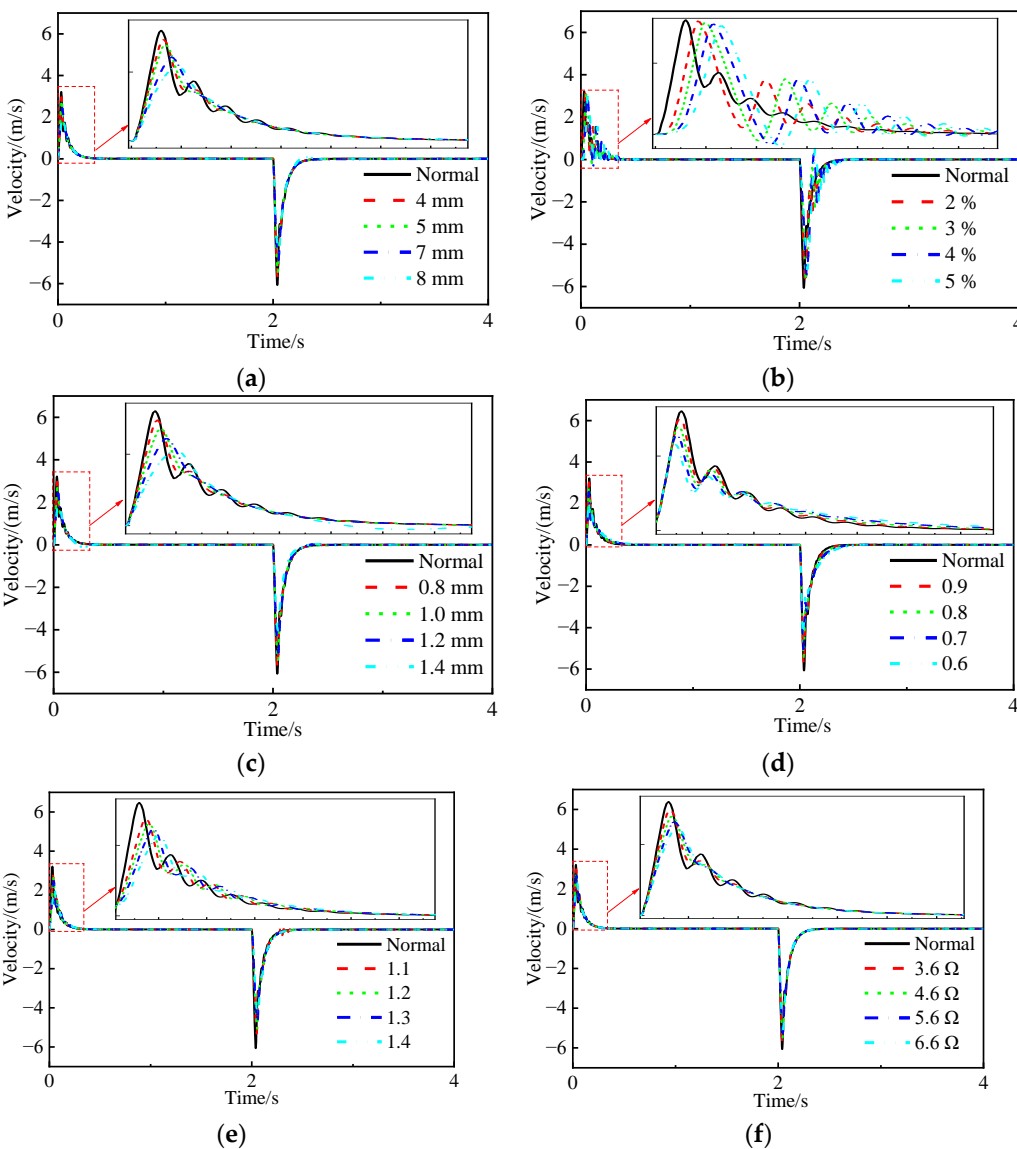

**Figure 12.** Velocity response of actuator cylinder under different faults: (**a**) Internal leakage of hydraulic pumps; (**b**) Oil mingled with air; (**c**) Internal leakage of actuator; (**d**) Friction increases; (**e**) Loss of sensor gain; (**f**) Resistance increases.

*5.3. Fault Sample Data Collection*

A total of six types of faults are simulated above. Four sets of fault curves are obtained for each fault category by adjusting the parameters, and each set of data curves contains seven curves characterizing the system performance. The simulation duration is 16 s, and the sampling frequency is set to 100 Hz, so that 1600 data points can be collected for each curve. Since the duration of each displacement command is 2 s, the length of each sample is set to 576, which considers the training efficiency while ensuring that enough valid information is covered. To divide the simulation curve of length 1600 into multiple samples of length 576, the data is enhanced by using a sliding window. A total of 256 samples can be obtained by setting the step size of the sliding window to 4 for fault data, and 1024 samples can be obtained by setting the step size to 1 for normal data. After the above processing, 7168 sample sets were obtained, each with a shape of 576 × 6. Table 2 shows one of the fault sample sets.

**Table 2.** Data of a fault sample set.

| Num | Displacement of Actuator | Velocity of Actuator | Pressure Difference of Pump | Flow of Plunger Pump | Rotation Speed of Motor | Outlet Pressure of Accumulator |
|------|------|------|------|------|------|------|
| 1 | 0.2008 | −0.0008 | −2.6600 | 20.3202 | 2423.4024 | −5.1040 |
| 2 | 0.2008 | −0.0008 | −2.6600 | 20.31987 | 2423.3584 | −5.1040 |
| ... ... | ... ... | ... ... | ... ... | ... ... | ... ... | ... ... |
| 206 | −0.2139 | 2.6553 | −0.1578 | 81.1545 | 9727.8676 | −2.2610 |
| 207 | −0.1836 | 3.3891 | −0.4929 | 71.17593 | 8900.0614 | −2.8220 |
| ... ... | ... ... | ... ... | ... ... | ... ... | ... ... |
| 576 | −0.1506 | 0.0006 | 1.9900 | −17.57166 | −2010.5476 | −3.2170 |

*5.4. GADF-Based Image Processing of Fault Sample Sets*

GADF is a feature transformation method that can encode a one-dimensional signal sequence into a two-dimensional map, which is generally used in image classification problems. The essence of GADF is dimensionality enhancement, which can fully exploit the topological feature information in the sample to ensure topological recognition. Therefore, we consider the GADF method for feature transformation. The one-dimensional EHA fault sample data is encoded as an image, and a deep learning model performs topological classification. GADF not only preserves the regularity of each feature in the fault data, but also provides other topological feature information, such as multiple relative relationships between nodes, which plays a vital role in the subsequent work of stable extraction of topological features [36–38].

The steps of feature transformation by GADF method are as follows [37]:

Step 1: Numerical normalization. Each fault sample set in 5.3 is composed of six one-dimensional signals $X = (x_1, x_2, \cdots x_{576})$, and each one-dimensional signal consists of $n$ timestamps $t$ and their corresponding values $x$. To prevent the inner product from being biased towards the maximum, the data $x(t)$ associated with each moment in the failure sample set is normalized.

$$\widetilde{x}(t) = \frac{x(t) - x_{\min}}{x_{\max} - x_{\min}} \tag{19}$$

$\widetilde{x}(t)$ is the normalized result of the parameter at time $t$, $x_{\min}$ and $x_{\max}$ are the maximum and minimum values of the parameter in the corresponding data set, respectively.

Step 2: Polar coordinate transformation. The normalized fault data can be converted to the polar coordinate system by converting the scaled values to the angle $\varphi$ and the timestamp $t$ to the radius $r$.

$$\begin{cases} \varphi_t = \arccos[\widetilde{x}(t)] \\ \quad r = \frac{t_i}{N} \end{cases} \tag{20}$$

where $t$ is the timestamp, $t_i = (1, 2, \cdots N)$, and $N = 576$ in this paper.

Step 3: Trigonometric transformation. After transforming the 1D fault data into the polar coordinates, we can easily exploit the angular perspective by considering the trigonometric difference between each point to identify the temporal correlation within different time intervals. The GADF are defined as follows:

$$GADF = \begin{bmatrix} \sin(\varphi_1 - \varphi_1) & \cdots & \sin(\varphi_1 - \varphi_n) \\ \sin(\varphi_2 - \varphi_1) & \cdots & \sin(\varphi_2 - \varphi_n) \\ \sin(\varphi_n - \varphi_1) & \cdots & \sin(\varphi_n - \varphi_n) \end{bmatrix} \tag{21}$$

The matrix elements are corresponded to grayscale to generate grayscale images, and the different grayscale values are corresponded to different colors to get the desired feature profiles. The strongly nonlinear data encoded into the image by the GADF matrix has great sparsity, which can eliminate redundant multimodal information, weaken the nonlinearity of the data, and reduce the noise [37], so the classification accuracy can be improved by using the GADF image as the input of CNN.

A total of 43,008 GADF images can be obtained by the GADF transformation of the fault sample sets. The size of each image is set to $64 \times 64$, and every six images constitute a sample set. Figure 13 shows the GADF images of some faults.

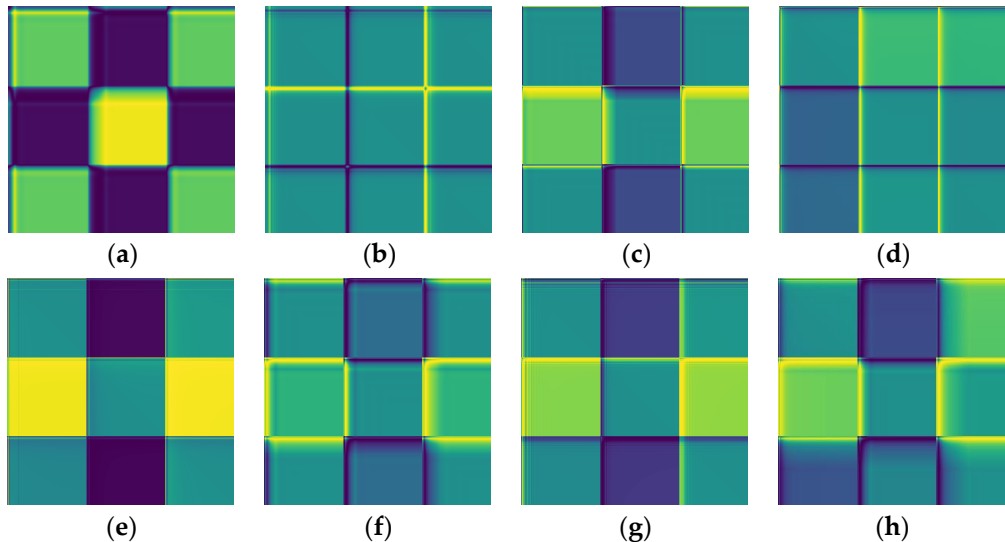

**Figure 13.** GADF images: (**a**) Internal leakage of hydraulic pumps (displacement of actuator); (**b**) Oil mingled with air (velocity of actuator); (**c**) Internal leakage of actuator (rotation speed of motor); (**d**) Friction increases (pressure difference of pump); (**e**) Loss of sensor gain (flow of plunger pump); (**f**) Oil mingled with air (rotation speed of motor); (**g**) Friction increases (outlet pressure of accumulator); (**h**) Internal leakage of hydraulic pumps (pressure difference of pump).

## 6. Typical Fault Diagnosis of EHA

In Section 5, 7168 sample sets were obtained by simulation experiments, and each sample set contains six fault data of length 576 and six GADF images of size $64 \times 64$. We divide these sample sets into 5012 training sets, 1435 testing sets, and 721 validation sets in the ratio of 7:2:1. Classification labels zero to six are added to each sample set, where zero represents normal, and one to six represents each of the six typical failure modes mentioned above. To verify the performance of the MSPSO-FCNN under noise interference, Gaussian white noise obeying $N(0, 0.5)$ is added to each feature of the original EHA fault data.

We use Tensorflow 2.0 to train the model, and the development environment is Py-Charm Community Edition 2021.3.3. The CPU is Intel(R) Core(TM) i7 9750H @2.6GHz, the running memory is 16 GB, the graphics card is NVIDIA Geforce GTX 1650, and the operating system is Windows 10.

We adopt the following metrics to evaluate the effectiveness of aircraft EHA troubleshooting [39]:

(1) Accuracy

The accuracy rate represents the ration of all samples with correct predictions. It is calculated as follows:

$$Acc = \frac{TP + TN}{TP + TN + FP + FN} \tag{22}$$

*TP* denotes the number of positive samples correctly classified, *TN* denotes the number of negative samples correctly classified, *FP* denotes the number of positive samples incorrectly classified, and *FN* denotes the number of negative samples incorrectly classified. In the multiclassification task, each category is considered a positive sample, while all other categories are considered negative samples.

(2) Precision and Recall

Precision is the ratio of correctly predicted positive classes to all items predicted to be positive. And recall is the ratio of correctly predicted positive classes to all items that are actually positive.

$$P = \frac{TP}{TP + FP} \tag{23}$$

$$R = \frac{TP}{TP + FN} \tag{24}$$

(3) $F_1$

$F_1$ is the harmonic mean of precision and recall, which is expressed as follows:

$$F_1 = \frac{2PR}{P + R} = \frac{2TP}{FP + FN + 2TP} \tag{25}$$

### 6.1. The Results of MSPSO Optimizes FCNN

The standard PSO algorithm and the MSPSO algorithm proposed in this paper are used to optimize the FCNN. The number of populations is set to 30, and the number of iterations is set to 50 in the experiments. We take the fault diagnosis accuracy of EHA as the fitness of particles. From Figure 14, we can see that the PSO and MSPSO fitness values increase as the number of iterations increases. The PSO converges at the 15th iteration with a global optimal fitness value of 0.8948. The MSPSO converges at the 40th iteration with a global optimal fitness value of 0.9756. It is not difficult to find that PSO obtains better fitness values than MSPSO when initializing the population due to the randomness of initialization. However, the standard PSO lacks the mechanism to jump from the local optimum and falls into the local optimum at the beginning of the iteration. Although the MSPSO algorithm also has stalled phases, it effectively escapes from the local optimal and achieves better results.

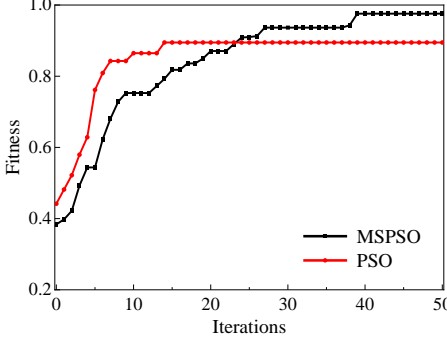

**Figure 14.** Fitness curve of PSO & MSPSO.

After the optimization of MSPSO, the optimal combination of FCNN hyperparameters are shown in Table 3.

**Table 3.** Optimal hyper-parameters of FCNN.

| Particle | Searching Range | Gbest | Particle | Searching Range | Gbest |
|----------|-----------------|-------|----------|-----------------|-------|
| $x_1$ | 4–64 (step: 2) | 22 | $x_{11}$ | Sigmoid, Tanh, Relu | Relu |
| $x_2$ | 4–64 (step: 2) | 28 | $x_{12}$ | Sigmoid, Tanh, Relu | Sigmoid |
| $x_3$ | 4–64 (step: 2) | 40 | $x_{13}$ | Sigmoid, Tanh, Relu | Sigmoid |
| $x_4$ | 4–64 (step: 2) | 18 | $x_{14}$ | 64–1024 (step: 2) | 482 |
| $x_5$ | 4–64 (step: 2) | 26 | $x_{15}$ | Sigmoid, Tanh, Relu | Sigmoid |
| $x_6$ | 2–8 (step: 1) | 3 | $x_{16}$ | Adam, Adagrad, SGD | Adam |
| $x_7$ | 2–8 (step: 1) | 3 | $x_{17}$ | 0.05–1 (step: 0.02) | 0.36 |
| $x_8$ | 2–8 (step: 1) | 2 | $x_{18}$ | 0.2–0.8 (step: 0.1) | 0.5 |
| $x_9$ | 4–20 (step: 1) | 11 | $x_{19}$ | 25, 40, 50, 100, 150 | 50 |
| $x_{10}$ | 4–20 (step: 1) | 7 | | | |
| | | | **Fitness: 0.9758** | | |

After optimization, the model parameters of 1DCNN and 2DCNN in the FCNN network are shown in Tables 4 and 5. 1DCNN has two convolutional layers, the first convolutional layer has 18 convolutional kernels and the second convolutional layer has 26 convolutional kernels; Moreover, the three convolutional layers of 2DCNN have 22, 28, and 40 convolutional kernels, respectively. Two sets of 1D feature vectors from 1DCNN and 2DCNN are stitched into a 5574 × 1 1D vector in the convergence layer, and a 482 × 1 feature vector is obtained after the Dropout layer and fully connected layer; finally, the Softmax layer classifies these feature vectors.

**Table 4.** The structural parameters of 1DCNN.

| Layers | Filter Size | Filter Number | Feature Size | Activation Function |
|--------|-------------|---------------|--------------|---------------------|
| Input layer | - | - | 576 × 1 × 6 | - |
| Convolutional layer 1 | 11 × 1 | 18 | 566 × 1 × 6 | Sigmoid |
| Pooling layer 1 | 2 × 1 | 18 | 283 × 1 | - |
| Convolutional layer 2 | 7 × 1 | 26 | 277 × 1 | Sigmoid |
| Pooling layer 2 | 2 × 1 | 26 | 139 × 1 | - |
| Fully connected layer | 1 × 1 | 3614 | 3614 | Sigmoid |

**Table 5.** The structural parameters of 2DCNN.

| Layers | Filter Size | Filter Number | Feature Size | Activation Function |
|--------|-------------|---------------|--------------|---------------------|
| Input layer | - | - | 64 × 64 × 6 | - |
| Convolutional layer 1 | 3 × 3 | 22 | 62 × 62 × 6 | Relu |
| Pooling layer 1 | 2 × 2 | 22 | 31 × 31 | - |
| Convolutional layer 2 | 3 × 3 | 28 | 29 × 29 | Relu |
| Pooling layer 2 | 2 × 2 | 28 | 15 × 15 | - |
| Convolutional layer 3 | 2 × 2 | 40 | 14 × 14 | Relu |
| Pooling layer 3 | 2 × 2 | 40 | 7 × 7 | - |
| Fully connected layer | 1 × 1 | 3614 | 1960 | Sigmoid |

### 6.2. Results and Discussion

The optimal FCNN network obtained in Section 6.1 is used to diagnose the typical faults of aircraft EHA, and ten repetitions of the experiment are conducted. The average result of ten repetitions is selected as the final fault diagnosis result of the MSPSO-FCNN fault diagnosis method for aircraft EHA. To illustrate that FCNN has a better fault diagnosis effectiveness than traditional 1DCNN and 2DCNN, we use the completed training 1DCNN and 2DCNN models for fault diagnosis of EHA. The test results are shown in Table 6.

**Table 6.** Diagnosis results of different models.

| Method | | Acc (%) | P (%) | F₁ (%) |
|---|---|---|---|---|
| Optimization Method | Basic Structure of CNN | $Acc$ (%) | $P$ (%) | $F_1$ (%) |
| PSO | 1DCNN | 79.51 | 79.78 | 79.57 |
| | 2DCNN | 84.60 | 83.41 | 83.93 |
| | FCNN | 89.20 | 84.69 | 86.82 |
| MSPSO | 1DCNN | 83.14 | 91.64 | 87.14 |
| | 2DCNN | 91.64 | 89.30 | 90.39 |
| | FCNN | 96.86 | 96.95 | 96.88 |

To intuitively show the classification effect of the proposed fault diagnosis method on the typical faults of EHA, the confusion matrix of the above six methods is given in Figure 15.

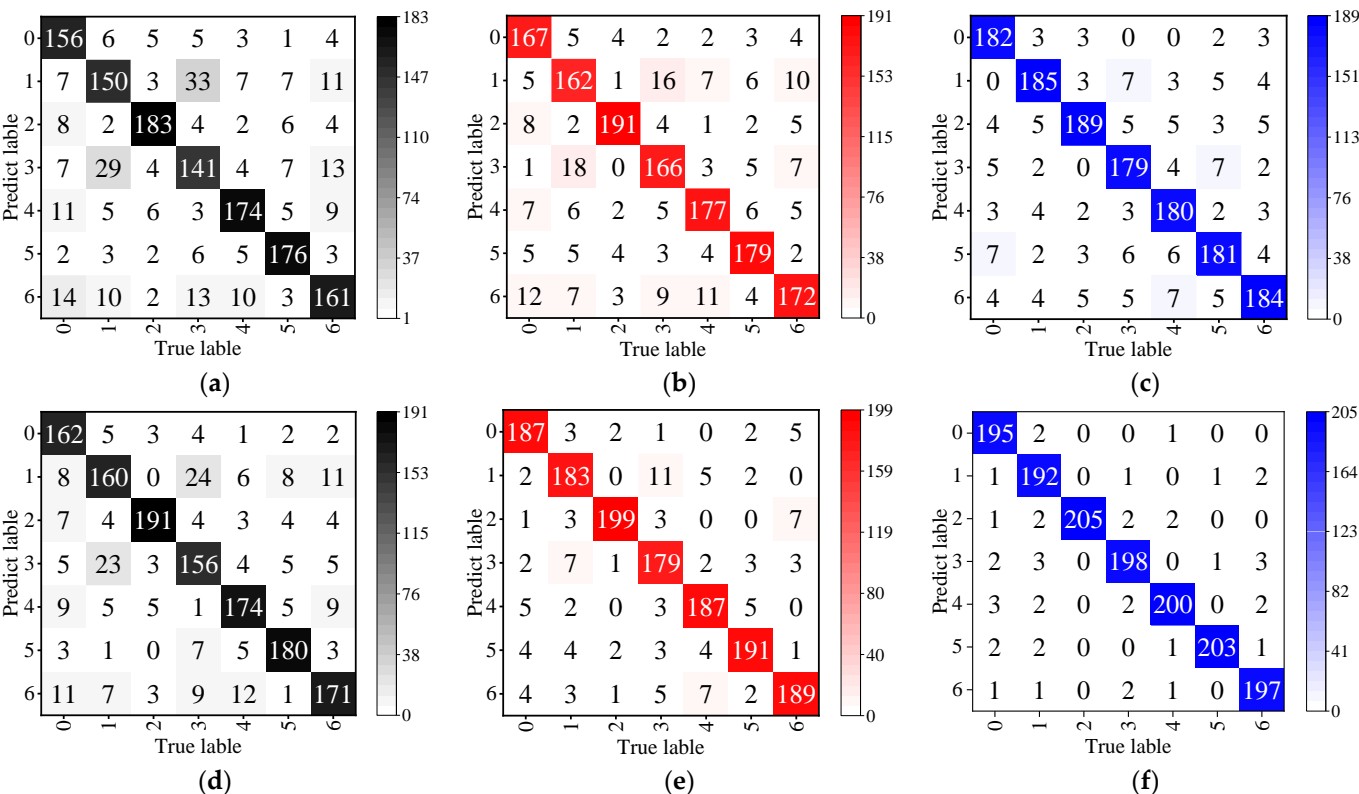

**Figure 15.** The confusion matrix of different models: (**a**) The confusion matrix of PSO-1DCNN; (**b**) The confusion matrix of PSO-2DCNN; (**c**) The confusion matrix of PSO-FCNN; (**d**) The confusion matrix of MSPSO-1DCNN; (**e**) The confusion matrix of MSPSO-2DCNN; (**f**) The confusion matrix of MSPSO-FCNN.

From Table 6, it is easy to find that MSPSO has better optimization results for all three types of convolutional neural networks compared with PSO. Among them, the accuracy of 1DCNN, 2DCNN, and FCNN after optimization of MSPSO is 4.6%, 8.3%, and 8.6% higher than that after optimization of PSO, respectively. The performance of MSPSO does not degrade due to the increase in optimization parameters, which reflects the great potential of the MSPSO algorithm for multi-objective optimization problems.

In addition, 1DCNN has poor recognition of fault 1 (78.1%), fault 3 (76.1%), and fault 6 (83.4%), especially for fault 1 and fault 3, which are two types of faults with relatively close characteristics. In Figure 15d, 23 sample sets belonging to fault 1 are classified as fault 3, and 24 sample sets belonging to fault 3 are classified as fault 1. But in 2DCNN, the recognition accuracy of fault 1, fault 3, and fault 6 was significantly improved, and the

average accuracy rate improved by 10.2% compared to 1DCNN. This is because, on the one-dimensional scale, the original fault signal cannot visually represent the fault characteristics of EHA, and some faults with close fault characteristics can be easily confused. However, the representation of two-dimensional image features is more abundant, and the GADF encoded image can eliminate part of the redundant information, making 2DCNN have certain robustness.

The FCNN method proposed in this paper improves the accuracy by another 5.7% compared to 2DCNN, reaching 96.86%, and it can effectively identify several types of faults, and the recognition rate of fault 2 is even 100%. This is because although 2DCNN can extract richer fault features from the GADF graph, it will lose some essential features that only exist in the one-dimensional original fault data. Nevertheless, FCNN can take advantage of the 2DCNN while retaining the original features in the 1D fault data, thus improving the fault diagnosis rate of aircraft EHA.

Furthermore, we tested the MSPSO-1DCNN, MSPSO-2DCNN and MSPSO-FCNN, respectively, with the original data without noise signal and obtained the results as shown in Table 7. Case 1 is the case without noise. It is easy to see from Table 7 that MSPSO-1DCNN is the most sensitive to noise, while benefiting from the sparsity of the GADF graph, MSPSO-2DCNN and MSPSO-FCNN have good anti-noise robustness.

**Table 7.** Robustness comparison of different models.

| Method | *Acc* (%) | |
|:---:|:---:|:---:|
| | **Cases 1** | **Cases 2** |
| MSPSO-1DCNN | 88.58 | 83.14 |
| MSPSO-2DCNN | 92.47 | 91.64 |
| MSPSO-FCNN | 97.21 | 96.86 |

*6.3. Comparison*

The proposed MSPSO-FCNN fault diagnosis model was compared with the widely used LeNet-5, AlexNet, GoogleNet, and GRU models in several metrics to verify the fault recognition effect further. The above models are trained ten times on the same training set, and the final average value was taken as the final result of each model, and the test results are shown in Table 8.

**Table 8.** Comparison between models.

| Model | *Acc* (%) | *P* (%) | *F*$_1$ (%) | Training Time (s) | Test Time (s) |
|:---:|:---:|:---:|:---:|:---:|:---:|
| MSPSO-LeNet-5 | 85.71 | 84.94 | 85.32 | 288.6 | 0.78 |
| MSPSO-GoogleNet | 95.47 | 91.98 | 93.69 | 573.2 | 1.11 |
| MSPSO-AlexNet | 91.64 | 91.29 | 91.46 | 449.5 | 0.98 |
| MSPSO-GRU | 84.94 | 89.96 | 87.38 | 350.6 | 0.65 |
| MSPSO-FCNN | 96.86 | 96.95 | 96.88 | 412.5 | 0.81 |

As seen in Table 8, the LeNet-5 and GRU models possessed lower recognition accuracy. Although GRU, as an NLP strategy, is very effective in processing one-dimensional data, the feature information of GRU input is not rich, and the robustness of GRU to noise signals is poor compared with other models with two-dimensional GADF diagram input. And, LeNet-5, as the simplest two-dimensional convolutional neural network structure with fewer layers, cannot extract fault features effectively. As for GoogleNet, and AlexNet, the recognition accuracy has been improved with the deepening of the model, in which GoogleNet's recognition accuracy reached 95.47%, which is comparable to the FCNN.

However, by comparing the differences in training and testing time between FCNN and GoogleNet, it can be found that the training time and testing of the GoogleNet network are 38.9% and 30.0%, respectively, higher than that of FCNN due to its complex structure and numerous parameters. Although the performance of GoogleNet is close to that of

FCNN, GoogleNet takes more time for training and testing, which affects the real-time fault diagnosis performance. Therefore, under a comprehensive comparison between multiple models, FCNN exhibits powerful feature learning capability and fault recognition performance and shows excellent test duration.

## 7. Conclusions

Combining the advantages of 1DCNN and 2DCNN, this paper proposes the FCNN model for typical fault diagnosis of aircraft EHA, which can extract richer fault features from GADF images while retaining the original features in one-dimensional fault data. To further improve the accuracy of fault diagnosis, this paper optimizes the structure of FCNN using the MSPSO algorithm, which combines four effective strategies: the initialization strategy based on homogenization and randomization, the adaptive inertia weights and learning factor strategies, the hybrid variation strategy, and the improved handling method of boundary-crossing particles. Through the process of fault diagnosis, the following conclusion can be reached:

(1) Compared to 1DCNN, 2DCNN with GADF map as input performs better in fault diagnosis of aircraft EHA. With the optimization of MSPSO, the 2DCNN can outperform the 1DCNN by up to 10.2% when diagnosing typical aircraft EHA faults.

(2) FCNN combines the advantages of 1DCNN and 2DCNN, and it can extract richer fault features from GADF images while retaining the original features in one-dimensional fault data. With the optimization of MSPSO, the accuracy of FCNN is 96.86%, which is 16.5% and 5.7% higher than 2DCNN and 1DCNN, respectively.

(3) The original data of aircraft EHA data are highly nonlinear, and encoding them into images by the GADF matrix can weaken the nonlinearity and reduce the noise, which makes 2DCNN and FCNN have better robustness.

(4) The algorithm quickly falls into the local optimum when using the standard PSO to optimize the CNN structure. In comparison, the MSPSO optimization algorithm proposed in the paper benefits from its integration of multiple strategies, can jump from the local optimum, and has better ergodicity. The accuracy of 1DCNN, 2DCNN, and FCNN under MSPSO optimization improved by 4.6%, 8.3%, and 8.6%, respectively, over PSO.

(5) Through comprehensive comparison with the LeNet-5, GoogleNet, AlexNet, and GRU models, the proposed MSPSO model possesses the highest test accuracy and shorter training time, which is very suitable for aircraft EHA fault diagnosis.

The proposed fault diagnosis method based on MSPSO-FCNN can effectively diagnose typical faults of aircraft EHA by only relying on the EHA state parameter information collected by sensors, without resorting to traditional complicated technical means or establishing an accurate analytical model of the system, which is superior to some traditional fault diagnosis methods. In the future, we will continue to apply this method to other fault diagnosis fields, such as aircraft engine fault diagnosis, aircraft landing gear faults, etc., and explore a fault diagnosis method with more expansive application fields based on MSPSO-FCNN.

**Supplementary Materials:** The following supporting information can be downloaded at: https://www.mdpi.com/article/10.3390/app12178562/s1, Table S1: Model parameters of EHA-FPVS.

**Author Contributions:** Manuscript Writing, X.L.; manuscript review and editing, Y.L. and Y.C.; simulation, X.L. and S.D.; project funding: Y.L.; formal analysis, X.L. and X.W.; reference and data collation, Z.Z. All authors have read and agreed to the published version of the manuscript.

**Funding:** This research was funded by the Aero-Science Fund of China, grant number (20200033052001) and Nanjing University of Aeronautics and Astronautics Postgraduate Innovation Base Open Fund, grant number (kfjj20200725).

**Institutional Review Board Statement:** Not applicable.

**Informed Consent Statement:** Not applicable.

**Data Availability Statement:** Not applicable.

**Conflicts of Interest:** The authors declare no conflict of interest.

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
