# Peer review of "Fault Diagnosis Method for Aircraft EHA Based on FCNN and MSPSO Hyperparameter Optimization"

_applsci, doi:10.3390/app12178562_

Round 1
Reviewer 1 Report
-The paper should be interesting ;;;
-it is a good idea to add a block diagram of the proposed research (step by step);;;;;;
-it is a good idea to add more photos of measurements (if any);;
-What is the result of the analysis?;;
-figures should have high quality;;;
-references to figures, tables in the text should be added;;;
-Fig 6 SI units should be added (if any);;
-please add photos of the application of the proposed research, 2-3 photos (if any) ;;;
-what will society have from the paper?;;
-Please compare the proposed method with other approaches/other methods;;
-why was the convolutional neural network (FCNN) used?
-references should be from the web of science 2020-2022 (50% of all references, 30 references at least);;;
-Conclusion: point out what have you done;;;;
Reviewer 2 Report
This study proposed a novel fault diagnosis method for aircraft EHAs based on an FCNN model that integrates both 1D- and 2D-CNN models. To obtain an optimal combination of FCNN hyper-parameters, this study adopts a MSPSO (Multi-strategy hybrid PSO) optimization algorithm.
The major comments are as follows:
1. Basic introductions about the structure of 1D- and 2D-CNNs, as well as fundamental PSOs, should be furtherly simplified.
2. Figure 16, along with related contents, is suggested to be arranged into an independent section named “Comparative Study”.
3. Note that the number of convolution layers and FC layers of both 1D- and 2D-CNNs are set as a fixed value of 2 in this study (see figure 1, 4, 5). Why not regard the numbers of conv-layers and FC layers as hyper parameters and optimize them with your MSPSO algorithm?
4. There are many outstanding types of ConvNets other than the 1D- and 2D-CNNs applied in this study, including LeNet, NiN, ResNet, GoogLE-Net, etc. However, the performance of your proposed model is not yet compared with such types of ConvNets. To make your results more convincing, such comparisons are necessary.
5. Moreover, based on comment 4, it seems to be a good idea to set the type of ConvNets (FCNN, LeNet, NiN, ResNet, GoogLE-Net) as a new hyperparameter, and optimize it with the aid of your MSPSO algorithm.
6. For 1-D fault data diagnosis, there are many NLP strategies that are proved to be effective (i.e. GRU, Deep LSTM, etc.). Therefore, comparisons between your CV-based GADF-FCNN model and various NLP models should also be taken into consideration. It is necessary to prove that your model performs better than NLP approaches on the EHA failure dataset.
7.'Remaining useful life prediction and predictive maintenance strategies for multi-state manufacturing systems considering functional dependence','Mission Reliability Evaluation for Fuzzy Multistate Manufacturing System Based on an Extended Stochastic Flow Network' and other newly published related studies from Applied Sciences should be added and discussed.
Reviewer 3 Report
To effectively identify the typical faults of the EHA, a fault diagnosis method based on the fusion convolutional neural networks (FCNN), which integrates the models of 1DCNN and 2DCNN, was proposed in this paper. Meanwhile, multi-stategy hybrid particle swarm optimization algorithm was used to optimize the hyperparameters of FCNN. This strategy of synthesizing merits of different methods/models to improve the accuracy is interesting and innovative. The proposed method would play an important role in the fields of fault diagnosis beside for EHA. Problems appeared in the paper are listed in the following,
1. In Eqs. (2)-(3), different symbols are suggested for the maximum-pooling and average-pooling at the left side of equations to avoid confusion.
2. The statement “The schematic diagram of the two-dimensional convolution operation is shown in Figure 3” contradicts with what presented in Figure 3.
3. Definition/explanation should be provided for GADF at its first appearance.
4. Figure 2 and its description should be carefully checked, especially the convolution kernel size and the output results.
5. In Figure 8, the block “Particle” should be placed correctly to avoid misunderstanding. Since it is fed back from MSPSO to FCNN after an iteration, “Particle” should connect to a judge block in MSPSO.
6. It is suggested to label the components in Figure 9 to better understand the work process of EHA-FPVS.
7. Convergence layer and Dropout layer of FCNN should be specified like other layers. In addition, as presented in Figure 8, there is a convergence layer to integrate 1D-CNN and 2D-CNN. The specific processes how to obtain this convergence layer in the analysis of EHA fault diagnosis should be provided.
8. Typos and grammatical errors: “accuracy improved by 10.2% over 1DCNN.” in the conclusion section, “...has performed well in recent years benefits from its sparsity...”, etc.
9. Correct format should be provided as referring to a literature, e.g., “Liu Jun ....”.
Round 2
Reviewer 1 Report
Figures should have better quality.
